Influence of the Great Amazon Reef System and Pleistocene sea-level drops on the phylogeography of Haemulon aurolineatum (Haemulidae)

Awhida-Robinson A. Karim 1 2
Torres-Hernández Eloísa 3
Pérez-Rodríguez Rodolfo 1
Piñeros Víctor Julio 4
Solís-Guzmán María Gloria 1
Angulo Arturo 5
Simões Nuno 6 7
Lasso-Alcalá Oscar M. 8 9
Monroy Mario 10
Domínguez-Domínguez Omar omar.dominguez@umich.mx 1
1 Laboratorio de Biología Acuática “J. Javier Alvarado Díaz”, Facultad de Biología, Universidad Michoacana de San Nicolás de Hidalgo , Morelia , Michoacán , Mexico
2 Programa Institucional de Maestría en Ciencias Biológicas, Facultad de Biología, Universidad Michoacana de San Nicolás de Hidalgo , Morelia , Michoacán , Mexico
3 Colección Nacional de Peces, Pabellón Nacional de la Biodiversidad, Departamento de Zoología, Instituto de Biología, Universidad Nacional Autónoma de México , Ciudad de México , Mexico
4 Departamento de Estudios para el Desarrollo Sustentable de Zonas Costeras, Centro Universitario de la Costa Sur, Universidad de Guadalajara , Cihuatlán , Jalisco , Mexico
5 Museo de Zoología/ Centro de Investigación en Biodiversidad y Ecología Tropical (CIBET) y Centro de Investigación en Ciencias del Mar y Limnología (CIMAR), Universidad de Costa Rica , San José , Costa Rica
6 Unidad Multidisciplinaria de Docencia e Investigación, Facultad de Ciencias, Universidad Nacional Autónoma de México , Sisal , Yucatán , Mexico
7 Laboratorio Nacional de Resiliencia Costera , Sisal , Yucatán , Mexico
8 Museo de Historia Natural La Salle, Fundación La Salle de Ciencias Naturales , Caracas , Venezuela
9 Green Earth Alliance , Miami , FL , United States of America
10 Laboratorio de Ecología Evolutiva, Departamento de Biología, Facultad de Ciencias, Universidad Nacional de Colombia , Sede Bogotá , Colombia
Jeffery Nicholas
Electronic publication date: 2025 Aug 11
Publication date: 2025
Volume: 13
Electronic Location ID: e19415
Received 2024 Nov 28; Accepted 2025 Apr 10
Copyright: ©2025 Awhida-Robinson et al.
Copyright year: 2025
Copyright holder: Awhida-Robinson et al.
License: This is an open access article distributed under the terms of the Creative Commons Attribution License, which permits unrestricted use, distribution, reproduction and adaptation in any medium and for any purpose provided that it is properly attributed. For attribution, the original author(s), title, publication source (PeerJ) and either DOI or URL of the article must be cited.
License URL: https://creativecommons.org/licenses/by/4.0/

Keywords: Genetic structure, Geographic barrier, Mesophotic reefs, Tomtate grunt, Greater Caribbean

Funding: Conahcyt 4019408 Universidad Michoacana de San Nicolás de Hidalgo CIC-UMSNH 2020-2024 CONABIO-LH003 CONABIO-NE018 Harte Charitable Foundation A. Karim Awhida-Robinson was supported by a MsC scholarship from the Conahcyt (Scholarship No. 4019408). This work was supported by funding provided for the Universidad Michoacana de San Nicolás de Hidalgo (CIC-UMSNH 2020-2024), CONABO grant number LH003. Field work was financed by the Harte Charitable Foundation through the Harte Research Institute TAMUCC, and CONABIO-NE018. The funders had no role in study design, data collection and analysis, decision to publish, or preparation of the manuscript.

==============================
Background

Phylogeography is based on the principle that species exhibit genetic structure shaped by biogeographic, ecological or environmental barriers, using both genetic and geographic components, offering valuable insights into evolutionary processes. In marine organisms, gene structure is influenced by life histories, geological events, and oceanographic conditions. The Greater Caribbean (GC), is a biogeographic region spanning from North Carolina, United States to northern Guyana, comprising three provinces: Northern, Central and Southern Caribbean. South of the GC is the Brazilian biogeographic province. Bellow the Amazonas-Orinoco plume the Great Amazon Reef System is present having mesophotic reefs situated beneath the freshwater discharge of the river. Each province is defined by distinct oceanographic conditions and habitat types, which play a significant role in shaping the evolutionary history of fish species. Due to its life history traits and the habitat heterogeneity across the GC, Haemulon aurolineatum, a widespread species found from Chesapeake Bay to Southern Brazil serves as an excellent model for studying evolutionary history of reef fishes in the GC region.

Methods

We use three nuclear DNA (nDNA) and one mitochondrial DNA (mtDNA) markers to study the phylogeographic history of H. aurolineatum. We performed gene structure, diversity indexes, haplotype networks, isolation by distance test, divergence time analysis and species delimitation methods in populations distributed through the geographic range of the species to understand the relation between the evolutionary history of the species, geophysical and biological aspects and make some taxonomic annotations.

Results

All four molecular markers revealed two distinct genetic groups: one predominantly distributed in the Northern province and the other mainly found in the Central, Southern Caribbean and Brazilian provinces. The divergence between these groups is estimated to have occurred around 800,000 years ago (Kya), this is attributed to the redirection of the Loop Current caused by climatic and oceanographic changes during the Pleistocene epoch. Our investigation has found genetic homogeneity among populations in the Central, Southern, and Brazilian provinces, which may be attributed to the ability of H. aurolineatum to migrate along the mesophotic reefs of the Great Amazon Reef System within the Amazonas-Orinoco plume region.

Introduction

Phylogeography is based on the idea that most species exhibit a degree of genetic structure related to biogeographic barriers, allowing inferences about evolutionary processes and patterns (Avise, 2000; Avise, 2009; Domínguez-Domínguez & Vázquez-Domínguez, 2009). In marine organisms, genetic structure is commonly influenced by the species’ life history traits, geological events and physical and environmental oceanographic conditions, such as biogeographic barriers (Sandoval-Huerta et al., 2019; Delrieu-Trottin et al., 2020; Araujo et al., 2022). In general terms, biogeographic barriers can be classified as hard and soft barriers (Cowman & Bellwood, 2013). Hard barriers refer to permanent or difficult-to-cross barriers; in marine organisms, these barriers can include land masses, such as the Isthmus of Panama; while soft barriers are permeable, typically caused by oceanographic and environmental conditions that can be crossed by some species (e.g., sandy gaps, ocean currents, and the discharge of major rivers) (Bowen et al., 2001; Cowman & Bellwood, 2013; Araujo et al., 2022).

Within the Western Tropical Atlantic (WTA), there is a region known as the Greater Caribbean (GC); this region extends from North Carolina in the United States (33°N) to northern Guyana (7°N) (Robertson & Cramer, 2014). Ecological discontinuities (e.g., differences in habitat) and environmental differences (e.g., transparency, temperature, salinity) are prevalent in this region and have influenced the evolutionary history of numerous fish species (Muss et al., 2001; Rocha et al., 2008a; Rocha et al., 2008b; Jackson et al., 2014; Piñeros & Gutiérrez-Rodríguez, 2017; Loera-Padilla et al., 2021). These environmental differences lead to the delimitation of three biogeographic provinces by Robertson & Cramer (2014). The Northern Caribbean province encompasses the entire Gulf of Mexico (GOM) and the Atlantic coast of Florida to North Carolina; the Central Caribbean province includes Central America and all oceanic islands; the Southern Caribbean province covers the entire continental coast of northern South America up to the northern border of Guyana. Each of these provinces is characterized by its unique habitats and environmental conditions such as transparency, temperature, productivity and salinity (Robertson & Cramer, 2014). Additionally, the Brazilian province which is within the WTA is characterized by having a variety of habitats such as eutrophic waters and some oceanic islands with oligotrophic waters, this region extends from the mouth of the Amazon River to Santa Catarina State (Floeter et al., 2008; Spalding et al., 2007; Pinacho-Pinacho et al., 2018).

The evolutionary history of fish species from the GC and surrounding regions have been influenced by different factors. These include the life history of species (e.g., Myripristis jacobus and Holocentrus adscensionis, Bowen et al., 2006; Acanthurus chirurgus, Rocha et al., 2002; Epinephelus striatus, Jackson et al., 2014; Sparisoma viride, Loera-Padilla et al., 2021), geological events (e.g., Acanthurus bahianus and Acanthurus coeruleus, Rocha et al., 2002; Abudefduf saxatilis, Piñeros & Gutiérrez-Rodríguez, 2017), ocean current circulation patterns (e.g., Agonostomus monticola, McMahan et al., 2012; Dormitator maculatus, Galván-Quesada et al., 2016 Awaous banana, McMahan et al., 2021), and the discharge of major rivers (e.g., Malacoctenus triangulatus, Dias et al., 2019; Selene setapinnis, Haemulon atlanticus, and Enneanectes altivelis, Araujo et al., 2022).

In the case of the Haemulidae family that encompasses 26 species within 10 genera in the GC (Robertson et al., 2023), its evolutionary history in the New World seems to be influenced by vicariant events due to geological changes (e.g., the emergence of the Isthmus of Panama and the Yucatán Peninsula), physical oceanographic conditions (e.g., the Amazon-Orinoco plume), as well as intrinsic characteristics of the species (e.g., multi-habitat species) (Rocha et al., 2008a; Rocha et al., 2008b; Bernardi et al., 2013; Villegas-Hernández et al., 2014; Palmerín-Serrano et al., 2021). On top of that, secondary contact of isolated populations due to long dispersal has been documented (Bernal et al., 2017). Within the Haemulidae family, the species Haemulon aurolineatum (Cuvier, 1830) has been shown genetic structure along its distribution range, despite its generalist ecological characteristics (Tavera, Acero & Wainwright, 2018; Araujo et al., 2022). It is found in subtropical and tropical latitudes, ranging from the state of Massachusetts in the United States, the Gulf of Mexico, the Caribbean Sea, the Guianas, as well as the north and south coast of Brazil, to the state of Santa Catarina, for more than 22,000 km of coastline (Lowe-McConnell, 1969; Castro-Aguirre, Pérez & Schmitter-Soto, 1999; Ribeiro et al., 2019; De Melo et al., 2020; Carvalho-Filho, 2023). It primarily inhabits coastal areas and is associated with reefs but can also occupy a diverse range of habitats, including seagrass beds, sand flats, patch reefs, natural hard bottoms, coral reefs, and artificial reefs over sandy and rough bottoms, at depths ranging from 1 to 140 m (Robertson et al., 2023). During its juvenile stage, it is also found in mangroves (Courtenay & Sahlman, 1978; Bravo, Eslava & González, 2009), in negative or hypersaline estuaries, such as coastal marine lagoons, near coral reefs and sandy bottoms with soft corals, where adults live (Cervigón, 1986; Cervigón, 2012; Tweedley et al., 2019). The species primarily feeds on small benthic invertebrates, plankton and algae (Carpenter & De Angelis, 2002). It produces pelagic eggs and larvae, although the duration of its pelagic larval stage remains unknown. For Haemulon flavolineatum, duration has been estimated at 15 days (Breder & Rosen, 1966; Robertson et al., 2023). According to this information, we expect to find structured populations of H. aurolineatum along its distribution range associated with soft barriers (e.g., sandy gaps, oceanic currents, major river discharges). To test this hypothesis, we use three nuclear (nDNA) and one mitochondrial (mtDNA) markers to investigate the phylogeographic patterns of H. aurolineatum and determine: (1) whether the species has gene structure between biogeographic provinces and (2) the influence of documented biogeographic barriers and life history traits on the species phylogeographic history.

Materials and Methods

Sample collections and data gathering

Sixty-five individuals of Haemulon aurolineatum were collected across the GC, with an additional 36 sequences downloaded from GenBank (http://www.ncbi.nlm.nih.gov/genbank/) and BOLDSYSTEMS (http://www.boldsystems.org/). In total, 101 individuals from 14 locations were included, covering the entire GC and representatives from each biogeographic province as defined by Robertson & Cramer (2014), as well as the Brazilian province (Fig. 1; Tables S1, S1.1 & S1.2).

Figure 1 Collection sites of Haemulon aurolineatum in the Greater Caribbean.

The biogeographic provinces of Robertson & Cramer (2014) are included in colour (see box). The Amazonian littoral, Amazonas-Orinoco barrier and the Brazilian province are also included. Sampling sites in the Northern Caribbean include US, North Carolina (EUC); US, Alabama (EUA); US, Florida (EUF); Mexico, Veracruz (MXV); Mexico, Campeche (MXC); Mexico, Yucatán (MXY). In the Central Caribbean: Mexico, Quintana Roo (MXQ); Belize, Riversdale (BLR); Jamaica, Falmouth (JAF); Puerto Rico, San Juan (PRS); Manzanillo, Costa Rica (CRM). In the Southern Caribbean: Colombia, Santa Marta (COL); Venezuela, Nueva Esparta (VZN). For the Brazilian province: Brazil, Bahía (BZB). Circles above localities represent haplotype proportion by haplogroups: green for Hg1 (Haplogroup 1) and blue for Hg2 (Haplogroup 2) for coxI. Map made on QGIS Development Team (2024). Photo credit: Dr. Omar Domínguez-Domínguez.

Most samples were obtained with the assistance of fishermen while others were collected using a pole spear. We took a fin clip from each individual and preserved it in absolute ethanol at −70 °C. Tissue samples and voucher specimens were deposited in the fish collection at the Universidad Michoacana de San Nicolás de Hidalgo (CPUM) and the Fish Collection at the Museo de Zoología of the Universidad de Costa Rica. Organism collection was supported and allowed by the following institutions and permits by Mexican Commission of Fisheries and Aquaculture under collection permits (CONAPESCA-PPF/DGOPA-/2013 and CONAPESCA-PPF/DGOPA-262/17); for Costa Rica collection permits R-SINAC-SE-DT-PI-029-2023 and the Comisión Institucional de Biodiversidad of the Universidad de Costa Rica (RESOLUCIÓN No. 377); and for Colombia, Universidad Nacional de Colombia, collection permits 76991-2024 ANLA-Unal-MinAmbiente and 3914-27.

Sequencing, alignment and substitution models

We extracted genomic DNA of the 65 samples collected using the Phenol-Chloroform protocol (Sambrook, Fritsch & Maniatis, 1989). We amplified fragments of one mitochondrial and three nuclear DNA markers: Cytochrome oxidase subunit 1 (coxI), ribosomal protein S7 (S7), rhodopsin (Rho) and myosin (Myh). PCR reactions were performed with a final volume of 12.5 µl (Table S2). The amplicons were purified with Exo-sap enzymes and sent for sequencing to Macrogen Inc., Seoul, South Korea. The sequences obtained for each molecular marker were visualized, edited, and manually aligned using MEGA v. 10.2.2 software (Tamura, Stecher & Kumar, 2021). To determine the best substitution model for each molecular marker, we used the program jModeltest v2.1.5 (Darriba et al., 2012) with the corrected Akaike Information Criterion (AIC). To detect and resolve heterozygous individuals, we used PHASE module from DnaSP v. 6 (Rozas et al., 2017). Analyses were conducted independently for each molecular marker. All sequences were deposited on GeneBank (Accession numbers: coxI: PQ571864 –PQ571924; S7: PQ588478 –PQ588521; Rho: PQ588522 –PQ588559; Myh: PQ588560 –PQ588601). The coxI mitochondrial marker was chosen due to the known variability in other population studies (Torres-Hernández et al., 2022; Bernal-Hernández et al., 2024; Torres-García et al., 2024) and the possibility to find a high number of sequences in the public repositories (e.g., Brazilian samples). The nuclear markers were chosen to try to have a high arrange of mutation rate, and the possibility to detect both, recent and ancestral events, as have been show in previous studies (Gaither et al., 2011; Sandoval-Huerta et al., 2019; Palmerín-Serrano et al., 2021).

Genetic diversity and haplotype networks

We calculated the number of haplotypes (Hn), haplotypic diversity (h), nucleotide diversity (π), and the number of polymorphic sites (S) of mtDNA and nDNA by biogeographic province in ARLEQUIN v. 3.5 (Excoffier & Lischer, 2010). In order to analyze the genealogical relationships between haplotypes and their geographical correspondence, we constructed a statistical parsimony network for each molecular marker using the software PopArt with the Median–Joining algorithm (Bandelt, Forster & Röhl, 1999).

Genetic structure and divergence time estimates

We conducted an analysis of molecular variance (AMOVA) to assess genetic differentiation. The groups were tested as follows: (1) a panmictic population (single genetic group), (2) by GC biogeographic provinces (Northern vs. Central vs. Southern Caribbean) (Robertson & Cramer, 2014) plus the Brazilian province (for coxI only), (3) the Northern Caribbean vs. Central, Southern Caribbean and Brazilian provinces (SCB-Provinces), and (4) between haplogroups obtained from the statistical parsimony haplotype network. Furthermore, we estimated population genetic differentiation by computing pairwise Fst values (1) among biogeographic provinces, (2) the Northern Caribbean compared to Central, Southern Caribbean and Brazilian province) and (3) between haplogroups obtained from the statistical parsimony haplotype network. Both analyses were performed using ARLEQUIN v. 3.5 (Excoffier & Lischer, 2010), with each analysis including 1,000 permutations to estimate significance. We also calculated uncorrected p-Distances (p-D) for all markers based on the same grouping scheme as the Fst values using MEGA v. 10.2.2 software (Tamura, Stecher & Kumar, 2021).

Divergence times were estimated in BEAST v. 1.8.4 software (Drummond & Rambaut, 2007) through a Bayesian inference phylogenetic reconstruction. To calibrate the molecular clock, we used seven species from the closely related genus Anisotremus, and four from Haemulon (Table S3) (Tavera, Acero & Wainwright, 2018). Due to the availability of gene markers in the outgroups, only the mtDNA marker coxI and nDNA marker S7 were included in the analysis. Fossil ages were sourced from the Paleobiology Database (https://paleobiodb.org/). The first calibration point was based on the split between Haemulon and Anisotremus estimated at 23–16 Mya (Million years ago) (mean: 0.2, SD: 0.8 offset: 16; calibration: Haemulidae) (Aguilera & De Aguilera, 2004). The second calibration was in the crown-group of Haemulon in the late Miocene: 11.6–7.24 Mya (mean: 0.9, SD: 0.4 offset: 7.24; calibration: Haemulon spp.) as reported by Gillette (1984). The third calibration point was in the crown-group of Anisotremus, with Pleistocene ages 5.33–3.6 Mya (mean: 0.2, SD: 0.1 offset: 3.6; calibration: Anisotremus spp.), as reported by Fitch & Lavenberg (1983). The analyses were run using an uncorrelated relaxed clock model with a lognormal Yule speciation process. Three independent MCMC runs were performed, each consisting of 10 million generations, sampling every 1,000 generations. Convergence of the chains was assessed by discarding the first 10% of generations as burn-in, using Tracer v 1.7 (Rambaut, Drummond & Suchard, 2018). Estimated parameters from the independent runs were pooled using the LogCombiner module in BEAST, and the maximum clade credibility tree was generated with the TreeAnnotator module. The analyses were executed on the CIPRES Science Gateway (Miller, Pfeiffer & Schwartz, 2010). Finally, the tree was visualized and edited on FigTree v 1.4.2 (Rambaut, 2014).

Isolation by distance

We estimated patterns of isolation by distance (IBD) using coxI gene throughout a Mantel test (Mantel, 1967) with 10,000 permutations in the “vegan” library on R (R Studio Team, 2022; Oksanen et al., 2022). To perform the correlation between genetic and geographic distance, we use linearized Fst values (Fst/1-Fst) (Rousset, 1997) as genetic distances, whereas the geographic distances between localities were obtained using the package “fossil” on R (Vavrek, 2011).

Species delimitation

We conducted two species delimitation analyses using four molecular markers. First, we inferred a species tree using BEAST v1.8.4 (Drummond, Rambaut & Suchard, 2016). For this approach, we designated Haemulon vittatum as the outgroup species (Tavera, Acero & Wainwright, 2018). The analysis employed an uncorrelated relaxed clock model with a normal Yule speciation process. The Markov chain Monte Carlo (MCMC) runs consisted of 30 million generations, sampling every 3,000 generations. To ensure convergence, we discarded the first 10% of generations as burn-in and assessed convergence using Tracer v1.7 (Rambaut, Drummond & Suchard, 2018). Second, we performed a Species Tree and Classification Estimation (STACEY v1.3.1; Jones, 2017) analysis using the STACEY template in BEAST v2.6 (Bouckaert et al., 2019). STACEY infers a “Species or Minimal Clusters” (SMC) tree under the birth-death-collapse tree prior, which does not require a guide tree (Petzold & Hassanin, 2020). This analysis was conducted using a strict clock model and a birth-death-collapse tree prior with the following parameters: CollapseHeight = 0.0001; CollapseWeight = 0.5; BirthDiffRate = 100; RelativeDeathRate = 0.5; OriginHeight = 100. The MCMC runs for this analysis also consisted of 30 million generations, with results saved every 30,000 generations. Convergence and ESS values were evaluated using Tracer v1.7 (Rambaut, Drummond & Suchard, 2018). The resulting files were processed in the Species Delimitation Analyser with a collapse height of 0.0001 and a similarity cut-off of 1.0, following the exclusion of the first 10% of generations as burn-in.

Results

Sequences and genetic diversity

For the mitochondrial gene coxI we obtained 97 sequences with a length of 628 base pairs (bp), exhibiting 43 polymorphic sites and 39 haplotypes. For S7, we obtained 44 sequences with a length of 387 bp, showing 46 polymorphic sites and 34 haplotypes. Regarding Rho, dataset contained 38 sequences with a length of 806 bp, featuring 17 polymorphic sites and 21 haplotypes. The dataset for the Myh gene consisted of 42 sequences with a length of 692 bp, displaying 34 polymorphic sites and 25 haplotypes. Samples from Brazilian province for the nDNA markers were not included in this study. The coxI gene exhibited the highest values of nucleotide diversity and haplotypic diversity in the Central Caribbean region and the lowest in the Brazilian province. The nuclear gene S7 showed the highest values of nucleotide diversity and haplotypic diversity in the Southern Caribbean region, and the lowest in the Northern Caribbean. In Rho, the highest values of nucleotide diversity and haplotypic diversity were found in the Southern Caribbean region, and the lowest in the Northern Caribbean. Lastly, in the Myh the highest haplotypic diversity was found in Southern Caribbean, the highest nucleotide diversity in Northern Caribbean and the lowest in the Central Caribbean (Table 1). We acknowledge that the absence of nDNA markers for the Brazilian province could introduce bias into our results. However, we believe the observed pattern of genetic homogeneity in the coxI Central and Southern Caribbean, as well as in the Brazilian province, is highly likely to persist even if nDNA markers are included in the analyses. This conclusion is supported by our own findings, where similar patterns of variation were observed in both mtDNA and nDNA data across the broader sampling area and provinces. Additionally, since mtDNA is known to exhibit greater variability than nDNA (see Vawter & Brown, 1986; Allio et al., 2017), we expect this pattern to hold. Nevertheless, we strongly recommend including nDNA markers in future studies to enhance the robustness of the findings.

Table 1 Mitochondrial and nuclear diversity indices by biogeographic provinces.

	Providence	N	Hn	h	π	S	
CoxI	NC	33	18	0.8400	0.00571	29	
CC	33	17	0.9010	0.00939	24	
SC	13	7	0.7310	0.00670	22	
BP	18	9	0.7060	0.00257	12	
S7	NC	12	9	0.9390	0.00924	14	
CC	12	9	0.9550	0.01429	13	
SC	20	16	0.9790	0.01590	19	
Rho	NC	12	2	0.1670	0.00041	2	
CC	6	11	0.8320	0.00543	12	
SC	20	8	0.8530	0.00528	9	
Myh	NC	12	8	0.8480	0.00490	14	
CC	8	5	0.8930	0.00484	8	
SC	22	12	0.9350	0.00366	12	
Notes.

Number of analysed individuals (N), number of haplotypes (Hn), haplotypic (h) and nucleotide diversity (π), and number of polymorphic sites (S). Biogeographic province as defined by Robertson & Cramer (2014): NC, Northern Caribbean province; CC, Central Caribbean province; SC, Southern Caribbean province and BP, Brazilian province.

Haplotype networks

The coxI haplotype network shows two haplogroups (Hg1 and Hg2) separated by eight mutation steps. Although Hg1 was composed of most individuals from the Northern Caribbean, also included five individuals from Central and one from Southern Caribbean, most of them showing peripheral position separated by three mutation steps from the nearest Northern Caribbean sample. Hg2 is composed by most of the individuals from the CSB-provinces, as well as two from Florida (St. Lucie County) and one from Yucatán (Sisal), mainly mixed with haplotypes of central province. The nDNA S7 marker shows more variation, with almost each sample forming their own haplotype without mixing, a somewhat geographic segregation is observed for the Northern Caribbean. The nDNA Rho and Myh genes show the haplotypes from the Northern Caribbean mostly segregated from those of the Southern and Central Caribbean samples (Fig. 2).

Figure 2 Median-Joining haplotype networks for the four genetic markers analysed.

(A) Mitochondrial marker coxI, and nuclear markers (B) S7, (C) Myh, and (D) Rho. Each circle in the network represents a distinct haplotype, and the circle size is proportional to the number of individuals sharing that haplotype. The colours within the circles correspond to different sampling locations, as indicated in the legend box. Small cross lines between haplotypes represent mutational steps, which indicate nucleotide differences between haplotypes. Number in with rectangle indicate mutation steps. The main haplogroups are highlighted in different colours to facilitate their identification: Haplogroup 1 (Hg1), corresponding to individuals from the Northern Caribbean, is shown in blue, while Haplogroup 2 (Hg2), including individuals from the Southern and Central Caribbean as well as the Brazilian province, is represented in orange.

Genetic structure and divergence time estimates

The results of the AMOVA for a single population found a high and significant ϕST, rejecting the null hypothesis of panmixia. The haplogroup arrangement (Hg1 vs. Hg2) maximized variation among groups for the coxI gene (Table 2). In contrast, for the nuclear gene S7 the highest variation was observed within populations. While Rho found the highest variation among populations within groups. Finally, for Myh, the variation was maximized within populations. Fst values as well as p-D show the same pattern as the haplotype networks in all molecular markers, where the Northern Caribbean is well differentiated in relation to the CSB-provinces (Fst = 0.58 to 0.80 (p < 0.05); p-D = 0.42 to 1.89; Table 3). The split of both, the Northern Caribbean and CSB-Provinces H. aurolineatum lineages was dated ca. 0.8 Mya (95% HPD: 0.35−1.38 Mya) (Fig. 3).

Table 2 Hierarchical analysis of molecular variance (AMOVA), using mtDNA and nDNA sequences.

Genetic groups analized	Source of variation	Variation (%)	Fixation index	P value	
coxI	Panmictic population (single genetic group)	Among groups	76.00	ϕST: 0.75996	0.00000	
Within populations	24.00			
Northern vs. Central vs. Southern vs. Brazilian	Among groups	27.52	ϕCT: 0.27525	0.09384	
Among populations within groups	49.73	ϕSC: 0.68619	0.00000	
Within populations	22.74	ϕST: 0.77256	0.00000	
Northern vs Central + Southern + Brazilian	Among groups	59.06	ϕCT: 0.59057	0.00000	
Among populations within groups	1.95	ϕSC: 0.04755	0.07331	
Within populations	39.00	ϕST: 0.61004	0.00000	
Haplogroup 1 vs Haplogroup 2	Among groups	79.17	ϕCT: 0.79168	0.00000	
Among populations within groups	5.56	ϕSC: 0.26679	0.00000	
Within populations	15.27	ϕST: 0.84726	0.00000	
S7	Panmictic population (single genetic group)	Among groups	29.11	ϕST: 0.29108	0.00000	
Within populations	70.89			
Northern vs. Central vs. Southern	Among groups	14.83	ϕCT: 0.14830	0.15445	
Among populations within groups	16.51	ϕSC: 0.19383	0.00391	
Within populations	68.66	ϕST: 0.31339	0.00000	
Northern vs. Central + Southern	Among groups	25.27	ϕCT: 0.25888	0.03812	
Among populations within groups	12.67	ϕSC: 0.17102	0.00098	
Within populations	61.44	ϕST: 0.38563	0.00000	
Hg1 vs. Hg2	Among groups	25.89	ϕCT: 0.25273	0.04203	
Among populations within groups	11.62	ϕSC: 0.15546	0.00098	
Within populations	63.11	ϕST: 0.36890	0.00000z	
Rho	Panmictic population (single genetic group)	Among groups	89.64	ϕST: 0.89640	0.00000	
Within populations	10.36			
Northern vs. Central vs. Southern	Among groups	16.92	ϕCT: 0.16920	0.66178	
Among populations within groups	106.06	ϕSC: 0.90707	0.00000	
Within populations	10.87	ϕST: 0.89135	0.00000	
Northern vs. Central + Southern	Among groups	9.15	ϕCT: 0.09146	0.23851	
Among populations within groups	80.96	ϕSC: 0.89109	0.00000	
Within populations	9.89	ϕST: 0.90105	0.00000	
Hg1 vs. Hg2	Among groups	41.11	ϕCT: 0.41107	0.05000	
Among populations within groups	50.19	ϕSC: 0.85227	0.00000	
Within populations	8.70	ϕST: 0.91300	0.00000	
Myh	Panmictic population (single genetic group)	Among groups	27.96	ϕST: 0.27958	0.00000	
Within populations	72.04			
Northern vs. Central vs. Southern	Among groups	8.84	ϕCT: 0.08842	0.72630	
Among populations within groups	35.76	ϕSC: 0.32856	0.00391	
Within populations	73.08	ϕST: 0.26920	0.00000	
Northern vs. Central + Southern	Among groups	9.44	ϕCT: 0.09435	0.13392	
Among populations within groups	21.55	ϕSC: 0.23798	0.00587	
Within populations	69.01	ϕST: 0.30988	0.00000	
Hg1 vs. Hg2	Among groups	12.32	ϕCT: 0.12323	0.35093	
Among populations within groups	20.17	ϕSC: 0.23006	0.01857	
Within populations	67.51	ϕST: 0.32494	0.00000	

Table 3 Fst values and p-distances.

Provinces of the GC	coxI	S7	Rho	Myh	
	F ST	p-D	F ST	p-D	F ST	p-D	F ST	p-D	
Northern-Central	0.516*	1.543	0.183*	1.716	0.461*	1.321	0.112	0.445	
Northern-Southern	0.655*	1.663	0.216*	1.701	0.228*	1.200	0.192*	0.436	
Central-Southern	0.001	0.796	0.039	0.394	0.179	0.518	0.005	0.330	
Northern-Brazilian	0.729*	1.616	n/d	n/d	n/d	n/d	n/d	n/d	
Central-Brazilian	0.078	0.650	n/d	n/d	n/d	n/d	n/d	n/d	
Southern-Brazilian	0.023	0.417	n/d	n/d	n/d	n/d	n/d	n/d	
Northern-Central & Southern (& Brazilian for coxI)	0.602*	1.619	0.301*	0.509	0.390*	0.505	0.199*	0.438	
HG1–HG2	0.803*	1.895	0.288*	0.582	0.583*	0.509	0.166*	0.418	
Notes.

Significant Fst values represented with asterisk* (α = 0.008 for coxI differences between provinces; α = 0.05 for all markers in NP-CP vs. SP-BP and HG1-HG2 comparisons; α = 0.016 for all ANDn molecular markers in provincial comparisons; alpha was calculated with Bonferroni correction). p-distance (p-D) values are presented as percentage (%). n/d no data available for the nuclear markers.

Figure 3 Time-calibrated species tree based on one mitochondrial gene (coxI) and one nuclear gene (S7).

White dots indicate the position of the three calibration points used. Purple bars represent the 95% highest posterior density intervals for divergence times. The numbers in the white boxes at each node represent the mean divergence time estimates. The numbers above the node correspond to the posterior probabilities obtained from the species tree. The number below the node indicates the posterior probability from the STACEY analysis. The blue clade represents the Northern Caribbean lineage, while the gold clade represents the Southern and Central Caribbean, as well as the Brazilian province lineage.

Isolation by distance

Mantel test shows a non-significant correlation between genetic and geographic distances (r = 0.0413; P = 0.3551).

Species delimitation

The species tree supported the separation of the Northern Caribbean from the Central and Southern Caribbean clusters with a posterior probability of one (Fig. 3). The results from STACEY for the global test, where we defined three minimal clusters (Northern vs. Central and Southern Caribbean vs. Outgroup), according to the hypothesis obtained in both phylogenetic reconstructions, showed high support (pp = 0.99).

Discussion

Our multi-locus study found two evolutionarily independent lineages inside the marine fish H. aurolineatum along its geographic distribution, corroborating previous studies (Tavera et al., 2012; Tavera, Acero & Wainwright, 2018; Tavera & Wainwright, 2019; Araujo et al., 2022). The evolutionary history of the species seems to be influenced by climatic oscillation during Pleistocene age and ecological complexities over the region.

Divergence of the two genetic groups

All the analyses conducted here reveal two well-supported genetic groups, one mainly distributed in the Northern Caribbean and the other in the CSB-Provinces, with genetic p-D of 1.89% and divergent event calculated to occur at 0.8 Mya. Nevertheless, previous studies in Haemulidae family provided older estimates (Tavera, Acero & Wainwright, 2018), placing the split between the two H. aurolineatum lineages at 2 Mya. The discrepancies between estimations could be associated with the used calibration points. Tavera, Acero & Wainwright (2018) used fossils of the stem lineage of Siganidae and Scatophagidae with an estimation date between 63.9 to 55.8 Mya, a stem lineage of the subfamily Acanthurinae with an estimation date between 57.3 to 50 Mya and a stem lineage of Luvaridae and Acanthuridae with an estimation date between 63.9 to 55.8 Mya. These three calibration points are concentrated in the Eocene-Paleocene epoch, which may bias the time estimates towards a more distant past (see: Magallón, 2010; Molak et al., 2012; Duchêne, Lanfear & Ho, 2014). On the contrary, in the present study we used three Haemulidae fossils, including the stem group of Haemulon and Anisotremus with dates between 23 to 16 Mya, the crown group of Anisotremus with dates between 5.3 to 3.6 Mya and the crown group of Haemulon with dates between 11.6 to 7.2 Mya (Palmerín-Serrano et al., 2021). These calibration points are better distributed over time, covering a window from 23 to 3.6 Mya. This broader time span in the calibration points enhances the accuracy of dating recent divergence events and helps mitigate temporal biases toward older estimates (Parham et al., 2012; Molak et al., 2012; Duchêne, Lanfear & Ho, 2014), as observed in the case of the two distinct genetic populations of H. aurolineatum. Consequently, we consider the dating findings in the present study to be more accurate than those reported by Tavera, Acero & Wainwright (2018).

Since IBD analysis indicates a non-significant correlation between genetic and geographic distance, the results suggest that the divergent event must be influenced by geological, oceanographical or biological-ecological barrier. Other fish species also show the isolation between Northern Caribbean (GOM) and CSB-Provinces samples, but they vary in divergences time, being calculated at 0.19 Mya in Dormitator maculatus (Galván-Quesada et al., 2016); or in genetic p-D, being 1.86% in Awaous banana (McMahan et al., 2021), to 5.3% in Gobiomorus dormitor (Guimarães Costa et al., 2017). In most of these studies, the genetic break was attributed to the effect of the Loop Current acting as a soft barrier. Loop Current is a warm water current that originates at the Yucatán Channel, flows through the strait between the Yucatán Peninsula and Cuba, loops back, and ultimately pass through the strait between Florida and Cuba (Vukovich, 1988; Oey, Lee & Schmitz Jr, 2003). This current could transport larvae from the Caribbean through the Yucatan Channel to the East, preventing its entrance to the GOM (Richards et al., 1993). This hypothesis is also supported by most of the samples (two of three) from the Atlantic side of Florida, which belong to the CSB-Populations haplogroup (Hg2), instead of the Northern Haplogroup (Hg1). Our divergence estimation (800,000 years ago) is in accordance with the dates estimated for the Günz Glaciation occurred between 850,000 to 600,000 years ago (Corrêa, 2021). The drastic climatic oscillations between repeated cycles of glaciation during the Pleistocene (Corrêa, 2021) dropped the sea level between 60 to 120 m lower than today, and coastlines extended horizontally between 10 and 100 kilometers, leading to the fragmentation and alteration of marine habitats (Jackson, 1992; Greenstein, Curran & Pandolfi, 1998; Ludt & Rocha, 2015; Diester-Haass, Billups & Lear, 2018).

The combination of the sea level drop (exposing large areas of the Florida and Yucatán peninsulas), with the dramatic sea surface cooling in the Gulf of Mexico (GOM), the prevailing north-easterly winds over the Caribbean, the strong outflow of the Mississippi River and the topography of the Yucatán Peninsula, restricted the Loop Current entry into the GOM. Instead, the current veered directly towards the Florida Straits and Cuba (Schmidt, Spero & Lea, 2004; Nürenberg et al., 2008; Arellano-Torres, Amezcua-Montiel & Casas-Ortiz, 2023) (Fig. 4). As a result, the GOM became isolated, leading to the genetic separation of two evolutionary lineages in H. aurolineatum. Nevertheless, a comprehensive explanation of genetic differentiation between GOM and CSB-Provinces populations must consider not only their origins but also their maintenance under present conditions. Moreover, if we take into consideration the mixture of haplotypes in the Caribbean of Mexico and Belize as well as Florida, high rates of gene flow over a few generations can eliminate genetic differentiation between populations. Therefore, it appears that the oceanographic barriers to dispersal or selection against migrants (or hybrids) are likely acting to maintain geographic variation currently.

Figure 4 Loop current configuration: actual and hypothetical during glacial periods.

Loop Current configuration: actual (black arrows) and hypothetical during glacial periods (orange arrows). Land extensions during maximum glaciation (green) and minimum glaciation (yellow) (map made on QGIS Development Team, 2024 with information from Ludt & Rocha (2015); Corrêa (2021)).

Homogeneity of the Central, Southern and Brazilian provinces

All our analyses across all genes show a pattern of genetic homogeneity in H. aurolineatum among the Central, Southern Caribbean and Brazilian provinces. Similar genetic homogeneity between the Central and Southern Caribbean provinces has been reported in other species, such as Abudefduf saxatilis, Sparisoma viride, Dormitator maculatus, Agonostomus monticola, and Awaous banana (McMahan et al., 2012; McMahan et al., 2021; Galván-Quesada et al., 2016; Piñeros & Gutiérrez-Rodríguez, 2017; Loera-Padilla et al., 2021). However, genetic homogeneity between the Southern-Central Caribbean and Brazilian provinces is an uncommon pattern in reef-dwelling fishes. Typically, this region displays genetic breaks, with the emergence of sister species pairs or populations among shallow-water reef fishes in the WTA, such as Ophioblennius macclurei and Ophioblennius trinitatis (Muss et al., 2001), Acanthurus bahanius (Rocha et al., 2002), Sparisoma frondosum—Sparisoma griseorubra (Robertson et al., 2006), Halichoeres radiatus—Halichoeres brasiliensis, Sparisoma axillare—Sparisoma rubripinne (Rocha et al., 2008a; Rocha et al., 2008b), Haemulon atlanticus, Coryphoptetus thrix, Enneanectes altivelis (Araujo et al., 2022), Centropomus irae (Carvalho-Filho et al., 2019)—Centropomus undecimalis (Malcher et al., 2023), among others. This commonly observed pattern of genetic differentiation is often attributed to the Amazon-Orinoco plume barrier, which assumed its current configuration during the early Pleistocene (∼2.4 Mya). Over time, the permeability of this barrier fluctuated with sea-level changes: it became less permeable during glacial periods, preventing species from crossing through the Great Amazon Reef System as the sea level dropped. In contrast, during interglacial periods, it became more permeable as the sea level rose, leading to the opening of the Great Amazon Reef System and allowing species to cross (Rocha, 2003; Ludt & Rocha, 2015; Araujo et al., 2022). Despite this, H. aurolineatum does not exhibit the typical genetic break seen in species like Acanthurus chirurgus (Rocha et al., 2002), Haemulon plumierii and Citharichthys spilopterus (De Jesus Gama-Maia et al., 2024).

To explain the observed genetic homogeneity of the tomtate grunt, we considered three possible scenarios. First, during interglacial periods, the increased sea levels may have made the Amazon-Orinoco plume barrier more permeable (Rocha, 2003; Araujo et al., 2022), allowing species like H. aurolineatum to cross, resulting in genetic homogeneity, as has been explained for Opisthonema oglinum (Ferreira-Araújo et al., 2024), Selene stepannis and H. aurolineatum (Araujo et al., 2022). Second, juveniles of H. aurolineatum, like other species in the Haemulidae family, may use mangrove and estuarine habitats during their life cycle (Castro-Aguirre, Pérez & Schmitter-Soto, 1999; Bravo, Eslava & González, 2009), potentially facilitating the exchange of individuals between the Greater Caribbean and Brazilian provinces. A similar pattern of connectivity across sandy and mangrove gaps has been observed in Anisotremus interruptus in the Tropical Eastern Pacific (Palmerín-Serrano et al., 2021). However, only two studies suggest that these species inhabit estuaries in their early stages, while most evidence points to H. aurolineatum being strictly marine (Darcy, 1983; Böhlke & Chaplin, 1993; Ornellas & Coutinho, 1998; Hoese & Moore, 1977; Robertson et al., 2023). This may be attributed to the findings of Castro-Aguirre, Pérez & Schmitter-Soto (1999) and Bravo, Eslava & González (2009), which report the presence of juvenile specimens in mangrove ecosystems within coastal lagoons, particularly in estuarine environments characterized by hypersalinity or negative salinity (Cervigón, 1986; Potter et al., 2010; Tweedley et al., 2019; Lasso-Alcalá et al., 2023). Thirdly, a continuous distribution and interchange of individuals through the Amazonian Coast plume may have occurred (including the Amazon-Orinoco plume zone). This region is characterized by large positive estuaries (Cervigón et al., 1992; Potter et al., 2010). This extends from the Orinoco River Delta (Venezuela) through the Guianas (Guyana, Suriname, and French Guiana) to the mouth of the Amazon River, and continues further east along the northern coast of Brazil, spanning throughout the states of Pará, Maranhão, and Piauí, up to the Parnaiba River Delta (see Fig. 1 and Fig. S1). This hypothesis is the most plausible, as data from the Global Biodiversity Information Facility (GBIF, 2024) shows a continuous distribution of the species (see Fig. S1). Notably, while H. aurolineatum is typically recorded in coastal reefs across its distribution range, in the Amazonian Coast region, the species has been recorded far from the coast and at more than 30–40 m deep (Lowe-McConnell, 1969; Collette & Rützler, 1977; Uyeno, Matsuura & Fujii, 1983; Guéguen, 2000; Moura et al., 2016; Pinheiro et al., 2018; Mrceniuk et al., 2021). These records from the Amazonian Coast region are aligned with a large area of reef patches known as the Great Amazon Reef System, which extends from eastern French Guiana to the north-eastern mouth of the Amazon River, off the coasts of Amapá, Pará, and part of Maranhão states in Brazil (Rosemary & McConnell, 1962; Collette & Rützler, 1977; Francini-Filho et al., 2018; Mrceniuk et al., 2021; Carneiro et al., 2022). This mesophotic reef (30–120 m depth), with its carbonate structures, acts as a corridor for genetic connectivity of reef-associated species across a wide range of depths (Cordeiro et al., 2015; Moura et al., 2016; De Mahiques et al., 2019; Banha et al., 2022; Carneiro et al., 2022). This reef habitat bridge appears to enable gene connectivity between H. aurolineatum populations in the Central-Southern Caribbean and Brazilian provinces, avoiding the outflow of the Amazon-Orinoco plume, as has been suggested for Coryphopterus venezuelae and species of the Halichoeres genus (Rocha, 2003; Volk et al., 2021).

Taxonomic implications

Phylogenetic studies of the Haemulidae family have revealed significant divergences within H. aurolineatum, identifying two distinct lineages (Tavera et al., 2012; Tavera, Acero & Wainwright, 2018). Taxonomically, two closely related species have been described: Haemulon aurolineatum, based on type specimens collected in San Domingo, Brazil (Cuvier, 1830), and Haemulon rimator, described from specimens collected in Charleston, South Carolina, and Pensacola, Florida (Jordan & Swain, 1884). Haemulon rimator was later considered a subspecies of H. aurolineatum by Ginsburg (1948) and subsequently synonymized by Courtenay (1961). However, Castro-Aguirre, Pérez & Schmitter-Soto (1999) treated it as a valid species. Most recently, Fricke, Eschmeyer & van der Laan (2024) again regarded H. rimator as a synonym of H. aurolineatum. Our population genetic analyses and species delimitation analyses confirm the presence of two well-differentiated lineages along the WTA, with all posterior probability values in the tree species delimitation analyses conducted of pp = 1−0.99, with p-D of 1.89% and a divergence event estimated to have occurred around 0.8 Mya. One lineage is predominantly distributed in the northern Caribbean, including samples near the known type locality of H. rimator, while the other is found mainly in the CSB-provinces, where H. aurolineatum was originally described (San Domingo, Brazil). According to Castro-Aguirre, Pérez & Schmitter-Soto (1999), these two species differ in the number of anal fin rays and coloration. They also noted differences in distribution, with H. aurolineatum ranging from Massachusetts to Brazil and H. rimator from Massachusetts to the Gulf of Mexico—findings that align closely with the distribution of the two lineages identified in this study. However, our species delimitation analyses must be taken with caution, since these methods are sensitive to sample size (Carstens et al., 2013). Accordingly, we recommend a comprehensive and integrative systematic analysis of H. aurolineatum across its distribution range, incorporating both morphological and genomic data at intra- and inter-regional spatial scales.

Conclusions

Our multi-locus study is the first to include samples from the entire distribution range of H. aurolineatum, covering all biogeographic provinces proposed by Robertson & Cramer (2014) and Brazilian province. This comprehensive approach provides valuable insights into the evolutionary history of H. aurolineatum across the WTA region. We identified two distinct lineages within this species: one primarily distributed in the Northern Caribbean and the other in the Central, Southern Caribbean and Brazilian provinces, with some genetic mixing occurring in the northern part of the Central Caribbean and Florida. These lineages likely diverged around 0.8 Mya, during Pleistocene, due to a combination of geological and oceanographic factors. Furthermore, our study highlights the genetic homogeneity observed among populations in the CSB-provinces, which may be facilitated by the dispersal of organisms through the Amazon-Orinoco plume, aided by the Great Amazonas Reef System acting as a corridor of gene connectivity. Taxonomically, the Northern lineage, previously proposed as a distinct species (H. rimator), has been synonymized with H. aurolineatum, which appears to correspond to the CSB-provinces lineage identified in this study. This highlights the need for a comprehensive integrative taxonomic study of what is currently recognized as H. aurolineatum. Additionally, we recommend the use of genomic data (RADseq) in order to assess the population patterns found in this study and elucidate the systematics of H. aurolineatum.

Supplemental Information

Supplemental Information 1 Supplementary information

Supplemental Information 2 S7 gene sequences

Supplemental Information 3 Rho gene sequences

Supplemental Information 4 Myh gene sequences

Supplemental Information 5 Cox1 sequences

We like to thank Mariana Raya Aguiar, Xavier Madrigal Guridi, Francisco Martínez Servín, Yareli Margarita López Arroyo, Aurora Lizeth Moreno Vázquez, Alfrancis Teresa Arredondo Chávez, Juan Antonio Sánchez Jiménez, Oscar Gabriel Ávila Morales, Sebastián Argüello, Georgina Palacios Morales, Rigoberto Moreno Mendoza and Quetzalli Hernández for field support. Patricia Jazmin Mayoral Loera, Adrian Emmanuel Uh Navarrete and Mohammed Hamed Ahsin Awhida for the technical assistance and translation review.

Additional Information and Declarations

Competing Interests

Author Contributions

Field Study Permissions

Data Availability

Oscar M. Lasso-Alcalá is employed by Green Earth Alliance.

A. Karim Awhida-Robinson conceived and designed the experiments, performed the experiments, analyzed the data, prepared figures and/or tables, authored or reviewed drafts of the article, and approved the final draft.

Eloísa Torres-Hernández conceived and designed the experiments, performed the experiments, analyzed the data, prepared figures and/or tables, authored or reviewed drafts of the article, and approved the final draft.

Rodolfo Pérez-Rodríguez conceived and designed the experiments, analyzed the data, authored or reviewed drafts of the article, and approved the final draft.

Víctor Julio Piñeros conceived and designed the experiments, prepared figures and/or tables, authored or reviewed drafts of the article, and approved the final draft.

María Gloria Solís-Guzmán conceived and designed the experiments, performed the experiments, authored or reviewed drafts of the article, and approved the final draft.

Arturo Angulo conceived and designed the experiments, authored or reviewed drafts of the article, and approved the final draft.

Nuno Simões conceived and designed the experiments, authored or reviewed drafts of the article, and approved the final draft.

Oscar M. Lasso-Alcalá conceived and designed the experiments, authored or reviewed drafts of the article, and approved the final draft.

Mario Monroy conceived and designed the experiments, authored or reviewed drafts of the article, and approved the final draft.

Omar Domínguez-Domínguez conceived and designed the experiments, authored or reviewed drafts of the article, and approved the final draft.

The following information was supplied relating to field study approvals (i.e., approving body and any reference numbers):

Comisión Nacional de Acuacultura y Pesca

Comisión Institucional de Biodiversidad de la Universidad de Costa Rica

Ministerio de medio ambiente, Colombia

The following information was supplied regarding data availability:

The sequences are available at GenBank: Cox1: PQ571864 –PQ571924; S7: PQ588478 –PQ588521; Rho: PQ588522 –PQ588559; Myh: PQ588560 –PQ588601.

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
