# Peer review of "Influence of the Great Amazon Reef System and Pleistocene sea-level drops on the phylogeography of Haemulon aurolineatum (Haemulidae)"

_PeerJ, doi:10.7717/peerj.19415_

## Round 0.1 · original submission · Major Revisions

Three reviewers were able to provide comments on this manuscript, and while they generally agree that the manuscript will be an important contribution to the conservation and phylogenetics of H. aurolineatum, some substantial revisions are needed. In particular, one reviewer notes the lack of nuclear sequences for Brazilian province individuals, potentially biasing the conclusion of genetic homogeneity in this region (it is also not clear to me why there is no nuclear data for this area).

Also of importance is some potential issues with the authors' hypotheses and conclusions, as noted by reviewer 2. Indeed, it would seem that during glaciation and consequently lower sea levels, the Amazon-Orinoco barrier would be less permeable to gene flow and lead to genetic isolation. These conclusions should be revisited and perhaps reworked, or rebutted with additional evidence.

I suggest the authors review and address each of the comments from these three reviewers, which will improve the analyses and conclusions of the manuscript.

Reviewer 1 ·

Basic reporting

no comment

Experimental design

The manuscript presents original primary research within the objectives and scope of the journal, addressing a relevant and significant research question related to the phylogeography of Haemulon aurolineatum. The study fills knowledge gaps by integrating genetic and biogeographic data to explore genetic connectivity and historical and oceanographic influences.

Validity of the findings

Validity of the Findings
The manuscript contributes to the understanding of the phylogeography of Haemulon aurolineatum by integrating genetic and biogeographic data. However, some areas need improvement to meet the journal's standards:

Data Robustness and Transparency:

The absence of nuclear marker data from the Brazilian province limits the robustness of the conclusions regarding genetic homogeneity. This limitation should be explicitly acknowledged, and its impact on the findings should be addressed in more detail.
The methods for processing the COX1 sequences (from 83 to 38 sequences used) lack sufficient detail. Clarifying this process is necessary to ensure transparency and replicability.
Statistical Soundness:

The manuscript could benefit from a deeper exploration of the non-significant correlation between genetic and geographic distances (IBD). Expanding on alternative explanations, such as local selection or ocean currents, would strengthen the statistical interpretation of the results.
Conclusions:

While the conclusions are generally aligned with the research question, they could be enhanced by providing clearer recommendations for future research. For instance, suggesting genomic approaches (e.g., RADseq) or ecological modeling to investigate mechanisms underlying genetic connectivity would add depth and novelty to the conclusions.
By addressing these points, the manuscript would better align with the standards for data robustness, statistical rigor, and comprehensive conclusions.

Additional comments

no comment

Annotated reviews are not available for download in order to protect the identity of reviewers who chose to remain anonymous.

Reviewer 2 ·

Basic reporting

The cited literature is appropriate and provides a relevant context for the study. However, the manuscript requires substantial textual improvements. Overall, some paragraphs are difficult to understand due to unclear structure, inconsistencies in terminology and abbreviation usage, and imprecise definitions.

Key points:

Lines 73-74: The definition of "hard barriers" is imprecise. It emphasizes the organisms' difficulty in crossing the barrier rather than the biogeographic barrier's characteristics.
Line 96: The Brazilian Province extends from the mouth of the Amazon River to Santa Catarina State (Floeter et al., 2008; Pinheiro et al., 2018).
Lines 98–111: This paragraph is convoluted, with excessive examples. It is challenging to determine the references for each example mentioned. In some cases, species are introduced correctly, while in others, genus abbreviations are used at first mention, which is incorrect. This inconsistency persists throughout the manuscript. Additionally, the overuse of parentheses impairs the paragraph’s readability.
Lines 187–191: AMOVA groupings are confusing.
Lines 284–285: Araujo et al. (2023) reported a result very similar to the one presented in this study. Although this study is cited several times, the authors fail to mention it in this specific context.
Figures 1, 2, and 3: It is recommended to modify the color scheme to be colorblind-friendly.
Figure 4: Excellent illustration!

Inconsistencies in terminology:

The title uses "Great Amazon Reef System," while line 449 refers to it as "Guayana-Amazonas Reef System."
Line 189: "Brazilian Province"; Line 190: "Brazil Province."
The simplification of the Caribbean province names creates confusion. They should consistently be referred to as "Northern Caribbean," "Central Caribbean," and "Southern Caribbean" throughout the text.

Time scale abbreviations such as Ma, Mya, and ya are used without definition. This inconsistency requires correction, along with similar issues throughout the manuscript.

Minor points:

Lines 94-95: Use "oceanic islands."
Line 109: Correct to Malacoctenus triangulatus, not "triangulates."
Lines 128–132: Add references.
Line 150: Specify "individuals of what?"
Lines 274 and 281: "The formation" is not appropriate terminology; consider rephrasing.
Line 289: Clarify if it is Acanthurinae or Acanthuridae.
Line 344: Correct to S. viride, not "viridae."
Line 350: The Caribbean and Brazilian species of Ophioblennius have long been formally recognized as O. macclurei and O. trinitatis, respectively.

Experimental design

Overall, the study is well-designed, employing appropriate methods and utilizing adequate and reliable data sources. While the research question has been vaguely addressed by previous studies, the present work provides a detailed phylogeographic analysis of Haemulon aurolineatum, addressing an important knowledge gap in reef fish ichthyology. The manuscript provides the necessary permissions for sampling. However, although the DNA sequences are included in the supplementary material, the text lacks any explicit statement indicating that the sequences have been deposited in GenBank, which constitutes standard practice for studies of this nature.

Validity of the findings

The study provides significant contributions to the understanding of the phylogeography of Haemulon aurolineatum. Notably, the hypothesis regarding the separation of the North Caribbean province lineage from the Central/South Caribbean and Brazilian provinces lineage appears well-conceived. However, the authors’ interpretation of the theoretical framework underlying their hypotheses on genetic homogeneity between the latter provinces raises concerns.

More specifically, based on the manuscript, the authors do not seem to fully understand the theoretical concepts related to the role of the Great Amazon Reef System, the Amazon-Orinoco Barrier, and Pleistocene sea-level fluctuations in shaping connectivity among reef fishes in the western Atlantic.
For instance, in lines 359–366, the authors state that the Amazon-Orinoco barrier “became more permeable during glacial periods, allowing species to cross as sea levels dropped, and less permeable during interglacial periods due to increased sedimentation and freshwater discharge (Rocha, 2003; Ludt & Rocha, 2015; Araujo et al., 2022).” A similar explanatory scenario is presented in lines 367–370.

While the consulted literature is appropriate, these studies suggest the opposite scenario: during glacial periods, the Amazon-Orinoco barrier becomes less permeable due to the "closure" of the Great Amazon Reef System as a genetic corridor, caused by low sea levels. Conversely, during interglacial periods, the Great Amazon Reef System may function as a genetic corridor below the freshwater plume, facilitated by high sea levels. This apparent discrepancy suggests a lack of mastery of the topic.

I recommend a significant revision of the manuscript, including structural adjustments and reconsideration of certain hypotheses.

Reviewer 3 ·

Basic reporting

The paper was well written and well described. The authors succeeded in using a multi-locus approach that covered the entire distribution range of Haemulon aurolineatum, showing valuable insights as the detection of two distinct lineages within this species: one primarily distributed in the Northern Caribbean and the other in the Central, Southern, and Brazilian provinces. They also brought important discussions regarding H. aurolineatum evolutionary history, as well as highlighted the need for a comprehensive integrative taxonomic study of what is currently recognized as H. aurolineatum. These results are robust findings, being a great contribution to the literature and the conservation and management of H. aurolineatum.

Experimental design

Just small doubts arose and may need to be considered:
Line 166. How many specimens in total were extracted? All the 64?
Line 167-169. Why were those markers chosen?

Validity of the findings

The conclusions were well stated.

---

## Round 0.2 · accepted · Accept

This manuscript provides a detailed analysis of the phylogeography of H. aurolineatum and understanding gene flow barriers leading to the observed genetic patterns. A reviewer and myself agree that the manuscript is now suitable for publication following the proofing stage. The authors have done an excellent job of responding to the reviewers' comments and incorporating the various suggestions, ensuring their data is available online as well.

Reviewer 2 ·

Basic reporting

I thought the authors were thorough in providing responses to comments and in applying the revisions on the manuscript text and figures.

Experimental design

Overall, the study is well-designed, employing appropriate methods and utilizing adequate and reliable data sources. While the research question has been vaguely addressed by previous studies, the present work provides a detailed phylogeographic analysis of Haemulon aurolineatum, addressing an important knowledge gap in reef fish ichthyology. The manuscript provides the necessary permissions for sampling. In this version, the authors provided the accession numbers of the sequences in Genbank.

Validity of the findings

I thought the authors were thorough in providing responses to comments and in applying the revisions on the manuscript text and figures.